# Towards Data-Driven Decision-Making in the Korean Film Industry: An XAI Model for Box Office Analysis Using Dimension Reduction, Clustering, and Classification

**DOI:** 10.3390/e25040571

**Published:** 2023-03-27

**Authors:** Subeen Leem, Jisong Oh, Dayeong So, Jihoon Moon

**Affiliations:** 1Department of Medical Science, Soonchunhyang University, Asan 31538, Republic of Korea; qlsl0519@sch.ac.kr; 2Department of AI and Big Data, Soonchunhyang University, Asan 31538, Republic of Korea; jso2562@sch.ac.kr; 3Department of ICT Convergence, Soonchunhyang University, Asan 31538, Republic of Korea; sodayeong@sch.ac.kr

**Keywords:** box office, classification, clustering, deep autoencoder, explainable artificial intelligence, machine learning, uniform manifold approximation and projection

## Abstract

The Korean film market has been rapidly growing, and the importance of explainable artificial intelligence (XAI) in the film industry is also increasing. In this highly competitive market, where producing a movie incurs substantial costs, it is crucial for film industry professionals to make informed decisions. To assist these professionals, we propose DRECE (short for Dimension REduction, Clustering, and classification for Explainable artificial intelligence), an XAI-powered box office classification and trend analysis model that provides valuable insights and data-driven decision-making opportunities for the Korean film industry. The DRECE framework starts with transforming multi-dimensional data into two dimensions through dimensionality reduction techniques, grouping similar data points through K-means clustering, and classifying movie clusters through machine-learning models. The XAI techniques used in the model make the decision-making process transparent, providing valuable insights for film industry professionals to improve the box office performance and maximize profits. With DRECE, the Korean film market can be understood in new and exciting ways, and decision-makers can make informed decisions to achieve success.

## 1. Introduction

Data-driven decision-making (DDDM) has become increasingly important in recent years due to the advancement of technology and the abundance of data available [1,2,3]. In many industries, people and organizations seek efficient and effective ways of processing and analyzing large volumes of data to support their decision-making efforts [4,5,6,7]. Traditional decision-making methods, which rely on personal knowledge, experience, and wisdom, are limited in dealing with big data effectively and are prone to bias and errors [8,9]. DDDM uses models and algorithms to process and analyze data sources for reliable decision support [10,11]. This approach has been widely applied in various industries, including medical diagnosis [12], financial risk prediction [13,14], public affairs governance [15], landslide susceptibility prediction [16,17], autonomous driving [18], and the safe operation of wastewater treatment processes [19,20], among others [21,22,23]. DDDM helps reduce the limitations of traditional decision-making methods, resulting in more accurate predictions and better decision support [1,6,9,11]. This capability of DDDM can result in the improved efficiency and reduced risks for various industries. Hence, developing new and improved DDDM models and algorithms has become a crucial research area and a significant trend in decision analysis.

The use of DDDM models in the film industry can also help ensure the industry’s steady growth [24,25]. By providing business decision support and guidance for filmmakers and distributors, DDDM models can help increase the chances of their project’s success, drive the industry’s continued growth, and ensure its long-term success [26]. For potential investors, using DDDM models can provide valuable information about the potential success of a film before they invest. DDDM models can help ensure the film industry’s steady growth and long-term success. For directors, DDDM models can provide insights into the success of their films and help them make informed decisions about which production companies to partner with, ultimately improving their chances of success. Hence, the use of DDDM models in the film industry has the potential to play a critical role in reducing the risks associated with the industry and ensuring its sustainable growth. By providing more accurate predictions and better decision-making support, DDDM models can help filmmakers, investors, and directors make informed decisions and improve the efficiency of their operations [27]. The use of DDDM models in the film industry is a clear example of how Industry 4.0 technology can revolutionize traditional industries and drive progress and innovation [26,27].

Predicting the success of a film at the box office is a complex task due to several factors [28]. The first factor is the short life cycle of films, which are typically screened within three months [29]. This short screening period, combined with the wide range of events that can occur during the screening, make it difficult to determine which marketing type will most impact the box office success. The second factor is the strong influence of word of mouth (WOM) on the film industry [30]. Unlike other industries, the demand for films is heavily impacted by what people say about the film to others. Despite these challenges, researchers have attempted to predict a film’s box office success using artificial intelligence (AI), namely data mining, machine learning (ML), and deep learning techniques [31,32]. These studies have focused on identifying the factors that impact the box office success and developing predictive models based on those factors. The studies suggest that big data, such as social media activity [33], Google searches [34], and Wikipedia page activity [35], can instantly predict a film’s box office success.

However, most studies have focused on the Western film market, and research on the Korean film market still needs to be explored. Our proposed model, DRECE (short for Dimension REduction, Clustering, and classification for Explainable AI), sets itself apart from previous research in several ways. Firstly, DRECE specifically targets the Korean film market, considering its rapid growth and the significant impact of the Korean Wave on the global entertainment industry [36,37]. Secondly, DRECE integrates XAI into the model, enabling the interpretation and understanding of factors affecting the box office and addressing the growing importance of explainable AI (XAI) in the film industry [38,39]. Thirdly, the framework starts by transforming multi-dimensional data into two dimensions through dimensionality reduction techniques, grouping similar data points using K-means clustering, and classifying movie clusters through ML models. By integrating these diverse techniques and building upon our previous work [40], our DRECE model provides a more comprehensive approach to analyzing the Korean film market, offering valuable insights and DDDM opportunities for film industry professionals to improve the box office performance and maximize profits. This approach represents a significant step towards DDDM in the Korean film industry through an XAI model for box office analysis using dimension reduction, clustering, and classification.

The main contributions of this study are summarized as follows:(1)We comprehensively understand the Korean film market using data collected from the Korean Film Council’s online integrated computer network. We offer a data visualization approach incorporating ML and data mining techniques to bridge the DDDM of Industry 4.0 and the film industry.(2)By considering various input variables representing movie characteristics, we identify the factors impacting a movie’s box office success in South Korea. Our proposed box office classification model is designed to assist film industry professionals in making data-driven decisions to increase the success of future films in the Korean market.(3)By reducing the feature dimension and applying data mining techniques, we effectively cluster movies and analyze box office trends for each cluster. Utilizing XAI, our model interprets the factors affecting the box office performance, providing valuable insights for decision-makers in the Korean entertainment industry to improve the box office success.

We organize the rest of the paper as follows: in Section 2, we list related studies on the box office. Section 3 describes our proposed approach, including the materials and methods used. In Section 4, we present the results of our experiments and engage in discussion. Finally, in Section 5, we conclude our findings and outline potential directions for future research.

## 2. Related Studies

Previous studies have attempted to predict the success of a movie at the box office by utilizing artificial intelligence techniques such as ML and deep learning. Table 1 highlights the utilization of AI techniques in these studies. The studies aim to identify the factors affecting the box office performance and develop predictive models. The findings suggest that significant data sources are crucial for predicting the box office success. The studies have employed different approaches, including ML algorithms and factors such as reviewing sentiments and social media data to predict the box office revenue. In this section, we review the related studies and their findings, highlighting the strengths and limitations of each approach.

Zhang et al. [41] conducted a study to predict the success of a movie at the box office before its theatrical release. To build the prediction model, they employed a multilayer backpropagation (MLBP) neural network (NN) with multiple inputs and outputs. The movies were divided into six categories, ranging from “blob” to “bomb,” based on their box office revenue. The input variables were selected based on market surveys, and their weight values were determined using statistical methods. The NN was optimized using theoretical guidance and experiments. A classifier with dynamic thresholds was used to standardize the output and improve the robustness of the model. A six-fold cross-validation experiment was used to measure the prediction model’s performance. The results showed that the MLBP model had a better prediction accuracy than the multilayer perceptron (MLP) method, with a 68.1% pinpoint accuracy and a 97.1% accuracy within one category. Kim et al. [42] proposed a novel method for predicting film box office earnings using social network service (SNS) data and ML algorithms. Three sequential forecasting models were developed to predict non-cumulative and cumulative box office earnings before, one week after, and two weeks after a film’s release. SNS mentions, weekly trends, and screening information were used as input variables. A genetic algorithm was used to select significant input variables, and three ML-based nonlinear regression algorithms were used to build the forecasting models. The results showed that using the SNS data and ML algorithms improved the accuracy of all three models. The research process involved selecting films, collecting screening and SNS data, determining the structure of the forecasting models, and selecting input variables. The conclusion was that their new approach, which used current screening and SNS information, improved the accuracy of forecasting box office earnings.

Hur et al. [43] proposed a new method that considered the review sentiment and employed non-linear ML algorithms for forecasting movie box office earnings. They used viewer sentiments from review texts as input variables in addition to conventional predictors and three ML-based algorithms, such as the classification and regression tree (CART), artificial neural network (ANN), and support vector regression (SVR), to capture the non-linear relationship between the box office and predictors. An independent subspace method (ISM) was applied to provide variable importance. The results showed that the proposed methods could make accurate and robust forecasts. A framework for box office forecasting was developed, and experiments were conducted to validate the ISM and verify the predictive performance of the proposed framework. The results showed that the ISM could assess variable importance robustly, and the proposed forecasting framework had a good predictive performance. Lee et al. [44] aimed to predict movie box office revenue using ensemble methods. The authors compared the prediction performance of decision trees (DTs), k-nearest neighbors (KNN), and linear regression using ensemble methods, such as random forests (RFs), bagging, and boosting, with a sample of 1439 movies. The results showed that the DTs using ensemble methods outperformed KNN and linear regression in predicting the box office revenue for the first, second, and third weeks after release. This study also compared the prediction performance between ensemble and non-ensemble methods within each algorithm and found that DTs using the ensemble methods provided better application effectiveness than KNN and linear regression analysis. The study was significant as it analyzed Korean movie data, which had rarely been investigated in the movie literature, and provided insights into predicting the movie box office revenue using ensemble methods.

Lee and Choeh [45] examined the relationship between movie resource powers and box office revenue, and how efficiency moderates the relationship between online word-of-mouth (eWOM) and revenue. Using data envelopment analysis, they found that movie efficiency had negative and positive moderating effects on different eWOM variables and their impact on the subsequent box office revenue. Their DTs, KNN, and linear regression analysis showed that movies with inefficient resources had better prediction performance than movies with efficient resources. The study added to the literature on eWOM by suggesting the production efficiency as a moderator between eWOM and the box office. The production efficiency produced by data envelopment analysis (DEA) still needed to be used in previous studies on the box office revenue. The authors showed that efficiency could affect the impact of different eWOM variables on the box office. Bogaert et al. [46] aimed to investigate the power of social media data (Facebook and Twitter) in predicting box office sales and determine which platform and data type are the most important. The authors used various prediction algorithms to compare models using movie, Facebook, and Twitter data. The analysis showed that social media data significantly improve the predictive power of traditional box office prediction models, with Facebook data performing better than Twitter data. The sensitivity analysis revealed that the volume and valence-based combination variables of Facebook comments were the most critical variables. The study found that Twitter had less of an impact on the box office sales due to the lower source credibility of Twitter users. The framework employed in the study was based on the cross-industry standard process for data mining (CRISP-DM) methodology. The study results are significant for practitioners, marketers, and academics who want to use social media data for box office sales predictions.

Pan [47] studied the factors affecting the box office revenue of the top 100 films in 2019. The results indicated that the score, potential audience, release schedule, and place of origin significantly impacted the box office revenue. The high fit of the model showed that these four independent variables well explained the dependent variable. The model’s residuals had weak autocorrelation due to low multicollinearity between the four variables. However, the positive comment rate was found to have no significant impact. The dummy variables revealed that films released on popular schedules had higher box office revenues, and films produced in China outperformed those produced abroad. The study found that film themes significantly impacted the box office revenue, with science-fiction films having the highest average box office revenue. The first-day box office was also found to significantly impact the total box office, reflecting a conformity effect among consumers. Li and Liu [48] researched predicting the box office revenue in the movie industry. They proposed an ML-based method for forecasting the box office revenue in the United States (US) and China. The method was tested and compared with eight other methods, and it was found that the support-vector-machine-based method using a gross domestic product (GDP) achieved the best results with a relative root mean squared error of 0.056 in the US and 0.183 in China. The results were validated using data from 2017, 2018, and 2019, and the mean relative absolute percentage errors were found to be 0.044 in the US and 0.066 in China. The study concluded that the proposed method effectively and efficiently predicted the nationwide box office revenue. The results provide evidence for the superiority of the support vector machine (SVM)-based method compared to other methods and demonstrate the potential of using economic factors in predicting the box office revenue.

Ni et al. [49] aimed to predict the box office revenue of films in China. The authors collected data from ENDATA including 5683 pieces of movie data and selected the top 2000 pieces for the prediction dataset. The authors used various types of Chinese microdata, a Baidu search index of movie names, and data on the coronavirus disease 2019 (COVID-19) epidemic to study the factors influencing the box office. Using a two-layer model architecture, they optimized the stacking algorithm using an ML technique. The base learners were extreme gradient boosting (XGBoost), light gradient boosting machine (LightGBM), categorical boosting (CatBoost), gradient boosting decision tree (GBDT), RF, and SVR. At the same time, the meta-learner was a multiple linear regression model. The prediction error was 14.49%, measured by the mean absolute percentage error. The results showed that the COVID-19 epidemic at the time of the movie’s release had a related impact on the movie’s box office. Velingkar et al. [50] studied the film industry, a multi-billion-dollar business significantly contributing to a country’s economy. They focused on the box office revenue of a movie, which is a crucial indicator of its popularity and can be influenced by various factors such as the production company, genre, budget, reviews, and ratings. The authors created an ML model that predicted a movie’s box office revenue based on the information available before its release to help investors make informed decisions. The authors used various algorithms, including XGBoost, RF, CatBoost, LightGBM, Ridge, and voting regressors, and considered factors such as the movie’s genre, original language, title, popularity rating, release date, budget, cast, crew, and more. The model considered the intended genre and target revenue and used the RF model to suggest a budget, runtime, star power, and expected popularity that would lead to the desired box office revenue.

Our research stands apart from prior studies by offering a unique perspective on the Korean film industry, focusing on classifying the box office types and analyzing factors contributing to the box office success. While earlier research primarily aimed to predict the box office revenue using various ML algorithms, our study employs a combination of ML techniques, XAI, dimension reduction, and clustering methods for a more comprehensive and explainable assessment of the film demand in the Korean market. By collecting data from the Korean Film Council’s online database, our study provides an in-depth understanding of the Korean film market, effectively clustering movies and analyzing the box office trends for each cluster. This clustering and analysis distinguish our work from previous studies that utilized single ML-based methods. Additionally, our study integrates established ML techniques with XAI, enabling a better interpretation and understanding of the factors affecting the box office performance. The valuable insights gained from our research can assist decision-makers in the Korean entertainment industry with making data-driven decisions to improve the box office success. In summary, our study’s unique contributions lie in its focus on the Korean film market, the integration of ML techniques with XAI for enhanced interpretability, and the combination of dimension reduction and clustering methods to assess and predict film demand more comprehensively.

## 3. Materials and Methods

### 3.1. Data Collection and Preprocessing

We gathered information about past box office performances through web crawling from the VOD Korea Box-office Information System (VKOBIS) [51], a computer system run by the Korean Film Council. The system quickly and accurately collects and processes movie-viewing data. Our data included the top 300 highest-grossing movies of all time, as shown in Table 2. The information for each movie included the title, production country, genre, director, actors, release date, and running time. If the data type was a character or category, it was initially collected in Korean and then translated into English by us. This information can be found in the Appendix A.

Each movie in the data had only one value for its production country and genre, but it could have zero or more than two values for its director and actors. Figure 1a,b compares the number of movies each country produces and their genres, respectively. Figure 1a shows that most movies were produced in South Korea or the U.S. The only movie made in the United Kingdom (U.K.) was “About Time,” the only movie made in Japan was “Your Name,” and the only movie made in France was “Taken 2.” In Figure 1b, action movies were the most prevalent, while war and family genres had only one movie each, “Harry Potter and the Sorcerer’s Stone” and “Operation Chromite.”

The title was excluded from the experiment, and the text variables “director” and “actor” were divided into separate values. The values were then labeled “CON” for the country, “GEN” for the genre, “DIRECT” for the director, and “ACT” for the actor. Finally, these characters and categorical variables were one-hot encoded, creating dummy variables by marking “1” if the corresponding value was present and “0” if not. The final experimental data had a total of 1028 features.

### 3.2. DRECE Model Construction

Figure 2 depicts our proposed DRECE framework. The framework starts by transforming multi-dimensional data into 2D data using two stages of dimensionality reduction techniques: deep autoencoder (DAE) and uniform manifold approximation and projection (UMAP). Then, K-means clustering is applied to the reduced data to group similar data points. The clustering result is added as a class label to the data, and one-hot encoding is performed on this class label. The proposed DRECE framework creates variables indicating which movies belong to which cluster. Finally, various ML models, such as logistic regression (LR), DT, RF, and CatBoost, are applied to classify the movie clusters, and the best-performing model is selected. The AI techniques that provide insights into the model’s decision-making process were used to make the model more explainable.

#### 3.2.1. Dimension Reduction

The reason for dimensionality reduction was that the high-dimensional data are challenging to visualize in their raw form and computationally demanding to process [52,53]. Therefore, by reducing the number of dimensions, the data can be presented in a more easily understandable format, and the computational load is reduced, making processing and analysis faster. As a result, dimensionality reduction can enhance the efficiency, accuracy, and interpretability of data analysis by focusing on the most significant features of the data [53]. We utilized the hidden variables of the DAE and UMAP to reduce the number of dimensions. To display the K-means clustering, we needed to shrink it to two dimensions.

DAE [54,55] is an NN architecture design for unsupervised learning. The goal of DAE is to reconstruct the input data. It consists of two main parts: an encoder and a decoder. The encoder transforms the input into a lower-dimensional representation, while the decoder transforms the lower-dimensional representation back into the original data. The idea behind DAE is that it can learn compact representations of data that are more meaningful than the raw input [32]. The complexity of the representation can be increased by adding multiple hidden layers to both the encoder and decoder parts of the network. 

The training of DAE involves minimizing the difference between the original input and the output of the decoder. We used the mean squared error (MSE) as the loss function for our DAE. The structure of the DAE is shown in Figure 3, and the details on the number of dense layers and the number of units in each layer can be found in the relevant case. We set the number of units in each layer to decrease or increase by a factor of 2. The activation function for the last layer in the decoder was sigmoid, while we used the rectified linear unit (ReLU) function to activate the other layers.

The latent variable in a DAE is a lower-dimensional representation of the input data created by the encoder. Using latent variables in a DAE means the encoder has learned a compact and meaningful representation of the input data. This low-dimensional representation eliminates fewer essential details and retains the most crucial information about the input. The decoder then utilizes the latent variables to reconstruct the original input, which should closely approximate the original data. As a result, latent variables become helpful for data compression, visualization, and feature extraction [54]. In other words, the latent variables of a DAE can be considered concise representations of the input data learned by the network during training [32,55]. 

Hence, we extracted these latent variables from the DAE and used them. We opted for a manifold learning technique to re-embed the data and aimed to learn the entire embedded manifold to optimize the clustering. While the autoencoder we used was a good choice for learning a meaningful data representation, it did not consider the local structure [56]. By combining the autoencoder with a manifold learning technique that considered the local structure, we could enhance the quality of the representation in terms of the clusterability. Therefore, the dimensionality of the latent space of the DAE was not immediately reduced to two dimensions; instead, a dimensionality reduction process using UMAP was performed separately.

UMAP [57] is a dimensionality reduction algorithm used to visualize high-dimensional data in a lower-dimensional space. UMAP combines Riemannian geometry, algebraic topology, and ML techniques to find a low-dimensional data representation that retains its structure. Unlike dimensionality reduction algorithms such as t-distributed stochastic neighbor embedding (t-SNE) [58] and principal component analysis (PCA) [59], UMAP can preserve the data’s local and global structure, and the algorithm can be adjusted through various hyperparameters, giving users greater control over the process [56,57]. Additionally, UMAP is less sensitive to changes in hyperparameters. Due to these features, UMAP is commonly used for data exploration and visualization. To provide a more intuitive and interpretable expression of the clustering results for the preprocessed data, we used DAE and UMAP to reduce the 1030-dimensional data to 2 dimensions (2D).

#### 3.2.2. Clustering Analysis

We used K-means clustering to cluster data points with similar characteristics in 2D data. K-means clustering [28,60] is an ML algorithm that divides a dataset into K clusters. The algorithm repeatedly updates the cluster assignments and cluster centroids until convergence is reached. The number of clusters, K, must be specified beforehand, and the algorithm’s goal is to minimize the sum of squares within the clusters. We used the elbow method [32,61] shown in Figure 4 to determine the optimal value of K. The elbow method involves comparing the sum of squares error (SSE) values for different numbers of clusters by plotting them on a graph. The optimal number of clusters is selected as the number corresponding to the point where the SSE value shows a steep decline followed by a gentle slope.

Nevertheless, alternative clustering methods, such as tclust, could be more suitable for our application. Tclust [62,63,64] is a generalization of K-means that can handle potential outliers and detect elliptical clusters. This method might provide more accurate results in cases where the data contain outliers, or the clusters have an elliptical shape, thus improving the overall performance of the clustering process. Tclust is typically implemented in R [65,66], while our study was conducted using Python, which led us to use the more general K-means clustering method. We recommend considering tclust or other clustering methods as alternatives to K-means for further analysis and improvements in similar applications, especially when using R or other programming languages that support tclust implementation.

#### 3.2.3. Movie Classification

We created four classification models to present the analysis results of the K-means clustering effectively. The original dataset, which consisted of 1030 input variables, posed a challenge as it needed labels or classes. Applying the supervised learning technique to the dataset made it difficult. 

To overcome this, we added a new variable by assigning a label of “1” to the box office belonging to each cluster and a label of “0” to the rest. Afterward, we utilized the 1028 variables and the cluster results (i.e., a label of “1” for the data belonging to a specific cluster and a label of “0” for data belonging to the other clusters) as input and output variables for each cluster to build four ML models, such as LR, DT, RF, and CatBoost. This was because they not only delivered high classification performance when the default values of the hyperparameters were set, or optimal values suggested in the previous studies were used, but they were also straightforward to interpret.

LR [67] is a statistical method used to model the relationship between a dependent variable and one or more independent variables. Given the independent variables’ values, it uses a logistic function to calculate a particular class’s probability. The logistic function generates a value between 0 and 1, representing the likelihood of the dependent variable being of a particular class. This probability value is then compared to a threshold, typically 0.5, to classify the dependent variable into one of two categories. LR can be used for binary classification tasks, where the dependent variable can only take two possible values. 

DT [45,68] is a tree-based model used for decision analysis and ML. It consists of internal nodes representing tests or conditions on the input features, branches representing the outcomes of these tests, and leaf nodes representing the final prediction. The DT algorithm works by splitting the data into smaller subgroups based on the values of the input features to find the splits that result in the highest accuracy. The final prediction is made based on the values of the input features at the leaf node, and DTs can be used for regression and classification tasks. They help understand the relationships between the input features and the target variable.

RF [50,69] is an ensemble learning algorithm that uses multiple DTs to make predictions. The algorithm works by training multiple DTs on different samples of data and different subsets of features. Each DT in the RF makes its prediction, and the final prediction is made by taking a majority vote among the predictions made by all trees. This process helps to reduce overfitting, increase accuracy, and make the model more interpretable. By using multiple DTs, RF also reduces the variance in the model, making it more stable. Based on the literature that suggests 128 trees is an appropriate number for the RF, we set the number of trees in our model to 128 [23].

CatBoost [23,70] is an advanced gradient-boosting algorithm that handles categorical variables effectively. It combines gradient boosting and categorical feature processing techniques to achieve better accuracy than other gradient boosting libraries. CatBoost makes it an ideal choice for datasets with many categorical variables. In addition, CatBoost has built-in mechanisms for handling overfitting and missing values, making it more robust and less prone to errors than other gradient-boosting algorithms. CatBoost is a powerful and flexible ensemble learning algorithm well-suited for many data analysis tasks. CatBoost is noted for delivering outstanding performance even with default hyperparameter settings, as noted by the authors, so we proceeded with the default values [70].

The above four models, LR, DT, RF, and CatBoost, are designed to be interpretable to provide insight into their decision-making processes. These four models are designed to be interpretable and easy to set up and are thus an accessible and practical choice for many classification tasks. Furthermore, the RF and CatBoost models are easy to interpret and offer high performance, making them an ideal choice for many classification tasks [23].

#### 3.2.4. Model Interpretation

The original dataset of 1028 input variables needed labeled or classified data, which posed a challenge for implementing supervised learning techniques. As a result, the interpretability of the RF and CatBoost models may have been weaker than that of the LR and DT models, which rely on labeled data to make predictions. To address this issue and increase the interpretability, we considered alternative techniques, such as Shapley values, which provide a more in-depth understanding of the model’s decision-making process.

Shapley additive explanations (SHAP) [71,72] is a method that provides explanations for the predictions made by ML models. It assigns a contribution value to each feature in the input data to explain the model’s predictions. The SHAP values are based on the concept of Shapley values from cooperative game theory and provide a way to fairly represent the contribution of each feature to the prediction, both in absolute terms and relative to other features. SHAP values help interpret complex models and understand the relationship between features and predictions. It helps explain the decisions made by the models, identify areas where they may be biased, and provide insight into the decision-making process. By calculating the SHAP values, we can understand which features are essential for predicting the movie type, making it a vital reference for understanding the proposed prediction model.

We considered using Tree SHAP for both the RF and CatBoost models to enhance the interpretability of our classification model further. Tree SHAP [23,39] is a method to interpret the output of tree-based ML models such as RF and CatBoost. It assigns contribution values to each feature in the input data to explain the predictions made by the model. By using Tree SHAP, we aimed to gain insights into the decision-making process and identify potential biases in the models. The results of the Tree SHAP analysis would be used to generate a list of essential features for predicting the movie type, which can serve as a reference for understanding how the proposed method makes predictions for the box office type prediction.

## 4. Experimental Results

### 4.1. Movie Data Compression and Clustering Analysis

We built a DAE model to condense the 1028 variables (excluding the movie name) from the original 1029 variables into a more compact representation of 16 dimensions. Figure 5 displays the loss of the DAE model during the training and testing phases. The loss was calculated using the MSE method. After 50 training cycles, the loss reached a minimum and stabilized, demonstrating that the DAE model successfully captured the essential features of the data and extracted the significant latent variables. Lastly, we utilized the UMAP algorithm to further reduce the 16-dimensional data into a 2-dimensional representation for easier visualization.

Figure 6 displays the results of applying these techniques to the data, showing the results of reducing the data to 2D using both DAE and UMAP, as well as the results of first reducing the data to 16D using DAE and then reducing it to 2D using UMAP. We configured the hyperparameters of UMAP to have an *n_neighbors* value of 8 and a *min_dist* value of 0.2. We found that when the data were reduced to 2D using DAE, only straight lines were displayed, which meant that DAE could not capture the non-linear structure of the data. In contrast, when we used UMAP to reduce the data to 2D, two large distributions were represented along the x-axis, corresponding to the two types of box office movies we were interested in identifying. However, it took work to distinguish between the two types of movies based on the UMAP plot.

To address this issue, we combined DAE and UMAP techniques by first reducing the data to 16D using DAE and then reducing it to 2D using UMAP. The combination of techniques was effective in identifying different types of box office movies. This is because DAE could capture the non-linear structure of the data, while UMAP could preserve both the local and global structure of the data. By using DAE first to reduce the data to a lower dimension and then applying UMAP, we achieved a more accurate and interpretable representation of the data. Hence, we found that combining the DAE and UMAP techniques was more effective in identifying different types of box office movies. The combination of techniques allowed for the capture of the non-linear structure of the data while also preserving the local and global structure of the data.

We used the elbow method to determine the optimal number of clusters. Figure 4 shows the SSE values for different numbers of clusters. We found that reducing the number of clusters from 1 to 6 resulted in a substantial decrease in the SSE. However, the decrease in the SSE was relatively minor for the number of clusters that was more significant than 6. Based on these findings, we concluded that this study’s optimal number of clusters was 6. The results of the data clustering, using the K-means clustering algorithm on the two-dimensional representation, can be seen in Figure 7a. Figure 7b shows the distribution of the box office data points across the different clusters.

### 4.2. Performance Comparison of Classification Models 

A confusion matrix, such as the one shown in Figure 8, is a table used to evaluate the performance of a binary or multi-class classifier. It provides a clear and concise summary of the classifier’s performance by displaying the number of correct and incorrect predictions in a clear format. The rows in the confusion matrix represent the actual class, while the columns represent the predicted class. Each cell in the matrix displays the number of observations corresponding to each combination of the actual and predicted classes. In a binary classification confusion matrix, the four most common items are: True positives (TP): The number of correctly predicted instances as positive.False positives (FP): The number of instances predicted as positive but are negative.True negatives (TN): The number of correctly predicted instances as negative.False negatives (FN): The number of instances predicted as negative but are positive.

When assessing how well a classifier works, several metrics are calculated using a confusion matrix. The confusion matrix is a table that shows the comparison between the classifier’s predicted results and the actual results for a set of data. Using a confusion matrix can also help improve the classifier’s performance. The following five metrics, calculated using Equations (1)–(6), are the best for understanding the classifier’s performance fully. A classifier is a tool to assign items to several predefined categories. The performance of a classifier is evaluated by comparing its predicted results to the actual results of a set of data.
*Accuracy* = (*TP* + *TN*)/(*TP* + *TN* + *FP* + *FN*)(1)
*Recall* = *TP*/(*TP* + *FN*)(2)
*F1-score* = *TP*/(*TP* + ½ × (*FP* + *FN*))(3)
*P_e_* = ((*TP* + *FN*) × (*TP* + *FP*) × (*TN* + *FN*) × (*TN* + *PF*))/*N*^2^(4)
*Kappa* = (*Accuracy* – *P_e_*)/1 – *P_e_*(5)
*MCC* = ((*TP* × *TN*) – (*FP* × *FN*))/√(*TP* + *FP*) × (*TP* + *FN*) × (*TN* + *FP*) × (*TN* + *FN*)(6)

Accuracy: Accuracy is a metric that measures the proportion of correctly classified instances by the classifier. It is calculated by dividing the number of correct classifications by the total number of instances in the data, as described in Equation (1). For example, if a classifier correctly classified 80 out of 100 movie instances, its accuracy would be 80%. However, in cases where the data are imbalanced, or false positive or negative results are costly, the accuracy may not be the best metric to use. For instance, if a classifier is designed to predict the box office hits, and it only correctly identifies 1 out of 10 movies as a hit, its accuracy would still be 10% even though it is not doing an excellent job of predicting the box office success.Recall: Recall, also known as the sensitivity or true positive rate, measures the proportion of box office hits correctly identified by the classifier in the dataset. It is calculated by dividing the number of box office hits correctly identified by the total number of hits in the dataset, as described in Equation (2). For example, if a classifier correctly identified 20 out of 25 box office hits, its recall would be 80%. The recall is an essential metric for evaluating the classifier’s ability to identify all box office hits, especially in cases where false negatives are costly. For instance, if a classifier is designed to predict which movies will be box office hits for a film production company, a high recall is crucial to ensure that all potentially successful movies are greenlit for production.F1-score: The F1-score is a metric that considers both the precision and recall, as described in Equation (3). It provides a balance between the precision and recall by considering metrics at the same time. The precision measures the ratio of the true positive box office hit predictions made by the classifier among all positive predictions. The recall measures the ratio of true positive box office hits among all actual positive box office hits in the data. The F1-score is the harmonic mean of the precision and recall and is calculated using a formula. An F1-score of 1 means perfect precision and recall, while 0 means the worst performance.Kappa: Cohen’s Kappa adjusts for chance agreement and evaluates the agreement between evaluators by considering both the observed agreement and the agreement expected by chance, as described in Equations (4) and (5). Kappa ranges from −1 to 1, with 1 indicating a complete agreement between the evaluators on the predicted box office hits and actual hits and a value less than 0 indicating an agreement worse than the chance. In other words, Kappa is a valuable metric for evaluating the performance of a classifier in cases where the data are imbalanced, and it considers both the observed agreement and disagreement in the predictions.MCC: Matthews correlation coefficient (MCC), as described in Equation (6), measures the quality of a binary classifier used to predict the box office types. It considers both the true and false positive and negative results. The MCC ranges from −1 to 1, with a value closer to 1 indicating a higher accuracy in the classifier’s predictions and closer to −1 indicating a lower accuracy. If a classifier’s predictions are random, its MCC value would be 0. The MCC is beneficial in cases where the dataset is uneven, as it considers both the accuracy and the ratios of true positive to false positive and true negative to false negative, making it a more reliable measure of performance in these cases.

We aimed to perform a feature analysis of the input variables most closely associated with different types of box office movies. To accomplish this, we constructed a classification model, which is an ML model that predicts the class or label of input variables based on their features or characteristics. However, we intended to use something other than this classification model for making future predictions. Instead, we trained the classification model on the entire dataset to identify the input variables that were most strongly related to each type of box office movie. We hoped to gain insights into the key factors that influence the success of different types of movies at the box office.

We used the entire dataset to evaluate the classification model rather than dividing it into training and test sets. Our decision allowed them to assess the model’s accuracy on the entire dataset and obtain a more comprehensive understanding of the relationship between the input variables and box office movie types. Furthermore, it was possible for the performance metric of the model to reach a value of 1 or close to 1 if the model was overfitting. Despite that, we constructed a classification model to perform a feature analysis on the input variables most closely associated with different types of box office movies. We trained the classification model on the entire dataset to gain insights into the key factors that influence the success of different types of movies at the box office. By using the entire dataset to evaluate the classification model, the authors were able to obtain a more comprehensive understanding of the relationship between the input variables and box office movie types.

Table 3 shows each cluster’s performance metrics and the study’s ML model. The random state for all the models was set to 42, and the decision tree’s maximum depth was set to 5 [68]. The logistic regression and decision tree models also performed well, with high accuracy, recall, F1-score, Kappa, and MCC values for many clusters. The logistic regression model even outperformed the other models for some clusters, achieving the highest accuracy for clusters A and D, the highest recall for cluster C, and the highest F1-score for clusters A, C, and D. Additionally, the logistic regression model had the highest Kappa and MCC values for clusters A, C, D, and F. These results suggest that the logistic regression model may be a good choice for identifying specific types of box office movies, particularly for clusters A, C, D, and F. However, it is essential to note that the optimal model choice may vary depending on the specific research question and dataset. 

Therefore, it is essential to carefully evaluate the performance of different models and consider their strengths and weaknesses before making a final selection. The RF model achieved perfect performance for all clusters. In Table 3, the RF model shows perfect classification accuracy, with all metrics equal to 1. The perfect classification accuracy can be explained by several factors, including the RF model’s inherent strengths and potential overfitting.

Inherent strengths of the RF model: As an ensemble model, RF builds multiple DTs and aggregates their predictions, reducing overfitting and improving generalization. Additionally, RF introduces randomness in the feature selection and bootstrapping samples, increasing the tree diversity and reducing the correlation between trees, resulting in a more robust and accurate model.Overfitting: The RF model may have to overfit the training data, resulting in perfect accuracy scores. Overfitting happens when the model learns the data’s noise rather than its patterns, leading to an exceptional performance on the training data but a poor performance on the new data. Evaluating the model on a separate validation or test set could be considered to check for overfitting.Entropy perspective: Entropy measures the impurity or randomness in the data. The RF’s trees use entropy to find the best-split points for each node. The perfect accuracy scores might result from the RF model’s ability to efficiently minimize the entropy at each split, producing highly accurate predictions.

Overall, the results suggest that the RF model was the most robust and accurate of the models tested, achieving perfect performance for all clusters. The other models performed well for some clusters but showed a lower performance for others, suggesting they may need to be more suitable for identifying all box office movie types.

### 4.3. Interpretability of Box-Office-Type Classification Model

Interpretability of the model refers to the ability to understand how it makes its predictions regarding the box-office-type classification. The interpretability of the model is important because it allows us to understand the strengths and weaknesses of the model and how it is likely to perform in real-world scenarios [73]. One way to interpret the model is to examine the model’s coefficients for each feature. The coefficients indicate the importance of each feature in the model’s predictions for the box-office-type classification. For example, if a particular director has a high coefficient, the model considers this director to be a strong predictor of the type of box office success the film may have.

The SHAP decision plot visualizes the contribution of each feature to a prediction made by an ML model for a specific instance or data point. The plot shows the relationship between the features and the prediction, with the essential features considered first. The features with higher absolute SHAP values had a more significant impact on the prediction. The plot displays the prediction as the mean prediction of the leaf nodes, with the leaf nodes’ size indicating each feature’s contribution. The interactions between features are represented by branches in the plot, with the branch’s size indicating the feature’s effect on the prediction. The value of the feature shows how much it contributes compared to the average contribution of all features for all instances. The SHAP decision plot provides an interpretable explanation for the relationship between the features and a model’s prediction.

The SHAP summary plot is a visual representation of the contribution of each feature to the prediction of a binary classification model. The plot displays the SHAP values for each feature, representing the feature’s impact on the model’s prediction. The x-axis displays the features, while the y-axis represents the contribution of each feature to the prediction, with positive values indicating a positive contribution and negative values indicating a negative contribution. The SHAP summary plot provides a clear view of the relationship between the features and the model’s prediction, making it an effective tool for understanding the behavior of binary classification models, such as movie genre classification or box office revenue prediction.

The SHAP violin plot is a graphical representation of the distribution of SHAP values for each feature in a binary classification model. The x-axis displays the features, and the y-axis shows the SHAP values split into two halves, one for the positive and one for the negative classes. The width of the plot at a point on the y-axis represents the density of the SHAP values for that feature, with the median being the center. The position of the plot on the y-axis indicates the direction of the feature’s contribution to the prediction, with positive contributions to the right of the baseline and negative contributions to the left. The width of the plot provides information on the variance in the contributions, with more expansive plots indicating a higher variance and narrower plots indicating a lower variance.

As shown in Figure A1, we analyzed Cluster A and revealed that it contained keywords such as “Yoo Ah-in” and “National Bankruptcy Day.” It was found that Yoo Ah-in, a famous Korean actor, was the most influential variable in this cluster. Furthermore, the other actors and directors from “Day of National Bankruptcy,” a movie where Yoo Ah-in played the main character, were also part of this cluster. Along with this movie, the cluster consisted of other directors or actors who have appeared alongside Yoo Ah-in in different movies. Interestingly, the French actor Vincent Cassel appeared in a movie with him. The article suggests that if Vincent Cassel and other non-Korean actors were to work with a Korean actor or director, it could lead to a good synergy effect. Combining various actors’ charms and acting skills and reflecting cultural diversity can produce more affluent and exciting work. The analysis of Cluster A could enhance the work’s perfection and attract the audience’s interest.

In Cluster B, shown in Figure A2, the main keywords were “Taken” and “animation.” The “Taken” series is a popular action movie with three installments, and the actors in the series are critical in this cluster. Liam Neeson, who played the main character in all three movies, was the most influential actor in the cluster. Many Korean directors, such as Lee Seok-hoon and Kwak Gyeong-taek, have produced successful films in Korea and are highly regarded in the industry. The cluster also contained many directors and voice actors who have produced American animations. If they participate in animation works produced in Korea, it can increase the possibility of gaining popularity worldwide. Since action movies and animations were distributed together in this cluster, new works can be created by producing action movies as animations or live-action animations. These initiatives can open new markets and create new revenue opportunities. Overall, this cluster provided insight into the potential collaborations between Korean and American actors, directors, and animators that could result in successful and globally famous works.

Cluster C was characterized by movies with suspenseful and action-packed plots, such as the famous “Maze Runner” and “Kingsman” series, as Figure A3 shows. These movies are known to captivate viewers with their thrilling action scenes and suspenseful developments, and the influence of the actors and directors who made them possible is significant. Notable Korean actors in this cluster include Kim Sang-ho, Baek Yun-sik, and Ma Dong-seok, known for their impressive performances in villainous roles, particularly in action scenes. Their inclusion in the cluster suggests the potential for them to be cast as the protagonists of new action movies within this cluster. In addition, Cluster C also included actors with exceptional singing abilities, such as Hugh Jackman and Cho Seung-woo. When combined with suitable directors or staff, this cluster showed promise for the involvement in film productions within the musical genre. Given the variety of styles within Cluster C, there is great potential for creating various types of action movies. However, new attempts and challenges will be required to achieve this, and these efforts could open new markets for the industry.

Cluster D was mainly influenced by middle-aged male Korean actors and actors who have appeared in the Avengers series, as shown in Figure A4. Notable actors in this cluster include Kim Seong-gyun, Ju Ji-hoon, Ko Chang-seok, and Lee Byung-hun. They could create a Korean version of the Avengers with their combined influence. Additionally, Korean actresses such as Kim Seong-ryeong and Han Hyo-joo had a significant presence in this cluster. If they were to take on roles similar to those played by Scarlett Johansson in The Avengers, they could attract public attention with their acting skills and charm. Furthermore, the actors in this cluster have the potential to participate in international activities, like the Avengers actors. As Korean actors become more involved in international film production, they could help elevate the Korean film industry to a global level. The analysis of Cluster D could lead to a more diverse and globally recognized film industry in Korea.

As depicted in Figure A5, Cluster E comprised actors who have shown their acting skills in films of various genres, including “Interstellar” and “Harry Potter,” which are representative works containing fantasy elements. The actors in this cluster are characterized by taking on diverse roles and displaying a wide range of acting skills, not limited to a single genre. Among the actors in this cluster, Jessica Chastain stands out as the most influential actress due to her outstanding performance in “Interstellar.” She has established herself as a performer who can perform in films of various genres and is highly recognized for her acting skills. As a result, the E cluster has the potential to create films of various genres by combining the appropriate actors. Recently, Jang Ki-yong and Lim Soo-jeong, actors from Cluster E, were selected for lead roles in a drama. These actors have shown consistent acting abilities in various works and are expected to receive high marks in dramas. Additionally, since there are many actors with acting abilities and charm regardless of the genre in this cluster, it is expected that the actors receiving attention from various works will appear in the future.

Figure A6 shows that Cluster F primarily comprised actors and directors who have contributed to the success of famous animated films such as Coco, Finding Dory, and Pororo. These movies have gained worldwide recognition and popularity, and it is believed that the technical abilities of the directors, the voice actors’ acting skills, and the use of visual effects all played essential roles in their success. On the other hand, Korean actors belonging to this cluster, such as Jang Dong-gun and Hyun Bin, are known for their versatility and are active in various fields, including advertising, dramas, and movies. Specifically, Dong-gun Jang, Hyunbin, Min-sik Choi, and Jiseong all share a history of appearing in historical drama films. Given the strengths of these actors, it is recommended that they appear as voice actors in animated movies or movies that heavily utilize computer-generated (CG) technology. This is because these actors can showcase their acting skills while also taking advantage of the impressive visual effects that CG technology can produce. Following these recommendations, we can expect Korean actors to create movies featuring innovative acting performances and stunning visual effects. 

## 5. Discussion and Conclusions

DDDM models utilize data analysis to inform decision-making in the film industry. These models analyze various factors related to a film’s potential success, such as the box office revenue and audience size, to predict its performance before its release. Filmmakers and investors can then use this information to reduce risk and make more informed decisions about the film’s production, marketing, and distribution. Data-driven decision-making models can potentially revolutionize the film industry by providing valuable insights and improving efficiency. By accurately predicting a film’s success, these models can help filmmakers and investors make better decisions about the resources they allocate toward producing and promoting a film. DDDM models, in turn, can lead to better-performing films and a more sustainable film industry overall. Predicting the success of a film can be challenging due to the complexity of the film industry and the various factors that impact a film’s success, such as the cultural and economic systems. However, data-driven decision-making models have the potential to help address these challenges, and provide more accurate predictions and better decision-making support for filmmakers and investors in the film industry.

To better understand the factors contributing to the success of box office movies, we analyzed the top 300 highest-grossing movies of all time. The data were collected using web crawling from the VKOBIS, a computer system run by the Korean Film Council, and included information on the title, production country, genre, director, actors, release date, and running time of each movie. We used the DRECE framework to process this data, which involved transforming multi-dimensional data into 2D data through dimensionality reduction techniques such as DAE and UMAP. The 2D data were then subjected to K-means clustering to group similar data points and classify the movie clusters. Finally, we applied ML models, including LR, DT, RF, and CatBoost, to classify the movie clusters. The results showed that the RF model performed best, with an accuracy and recall of 1.00 and an F1-score, Kappa, and MCC of 1.00. 

Although CatBoost is known for its excellent performance even with default hyperparameter values, the results in Table 3 did not meet our expectations. As a result, we decided to perform a grid search to find the optimal hyperparameters for CatBoost, including the learning rate, maximum depth of each tree, and the coefficient at the L2 regularization term of the cost function [74]. After specifying a wide range of hyperparameter values and running the grid search, the researchers achieved perfect performance for all clusters, with an accuracy, recall, F1-score, Kappa, and MCC equal to 1. However, it is worth noting that while this approach led to a perfect performance, it came with a high computing cost. Specifying a wide range of hyperparameter values and running a grid search can be computationally intensive and time-consuming, especially for large datasets. Therefore, this approach may only sometimes be feasible or practical in real-world applications, and trade-offs between the performance and computational efficiency must be considered.

Our study provides practical guidance for filmmakers seeking to maximize their chances of producing a hit by utilizing entropy-based methods in classification with a DT-based RF approach. However, it is essential to acknowledge the limitations of our proposed approach, which requires careful consideration of various factors such as the number and type of features or the quality and quantity of the data. In information theory, entropy is a measure of the uncertainty or randomness of a system. It represents the information needed to describe or predict the system’s state. Our DT-based RF algorithm used entropy to identify the most informative features and create a hierarchy of nodes representing increasingly specific rules for classifying instances. DT-based RFs can generate powerful models for predicting the box office success by recursively applying entropy-based splitting criteria.

Our findings demonstrate the potential of entropy-based methods to improve the classification performance of models by minimizing the uncertainty and maximizing information gain. By partitioning data based on the entropy, we can effectively identify unique clusters of movies with distinct strengths and characteristics that offer valuable insights for innovation and success in the Korean film industry. Furthermore, our paper significantly contributes to entropy-related issues in film industry analysis, providing a pioneering framework for future studies to build upon. Our work will inspire further research, leading to more advanced and sophisticated methods for analyzing film industry data. It is essential to acknowledge our approach’s limitations and conduct further research to enhance its generalizability and robustness when applying it to other datasets or contexts.

Unfortunately, the impact of COVID-19 on recent movie box office performance could not be studied, and Korean and Hollywood’s films were excluded from the analysis. Furthermore, a comprehensive analysis of the movie reviews on Korean portals and the differences in the box office performance between successful and unsuccessful films were not examined. However, our model provides valuable insights for decision-makers in the film industry to make data-driven decisions and improve future film success in the Korean market. Future research is necessary to thoroughly analyze the impact of COVID-19 on the box office performance of recent movies and to perform a more comprehensive and integrated analysis of the global film industry. Advanced natural language processing techniques will likely be utilized to provide a more systematic analysis, and the plan is to build an integrated platform that covers the worldwide film industry. As a result, it is recommended to leave these topics for future research, which will include a planned analysis of Hollywood data as well.

## Figures and Tables

**Figure 1 entropy-25-00571-f001:**
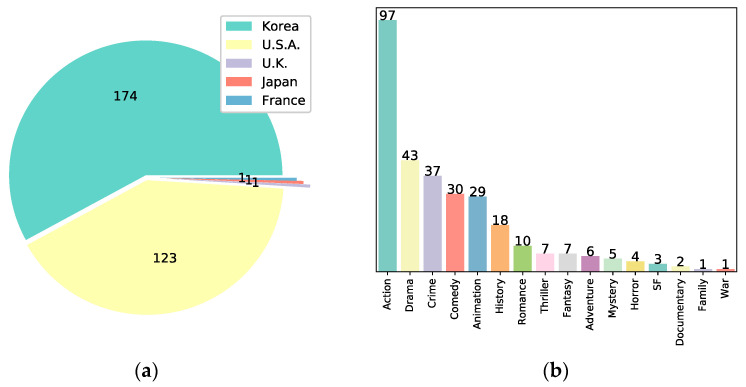
Information on box office movies. (**a**) Pie chart for the country; (**b**) Bar chart for the genre.

**Figure 2 entropy-25-00571-f002:**
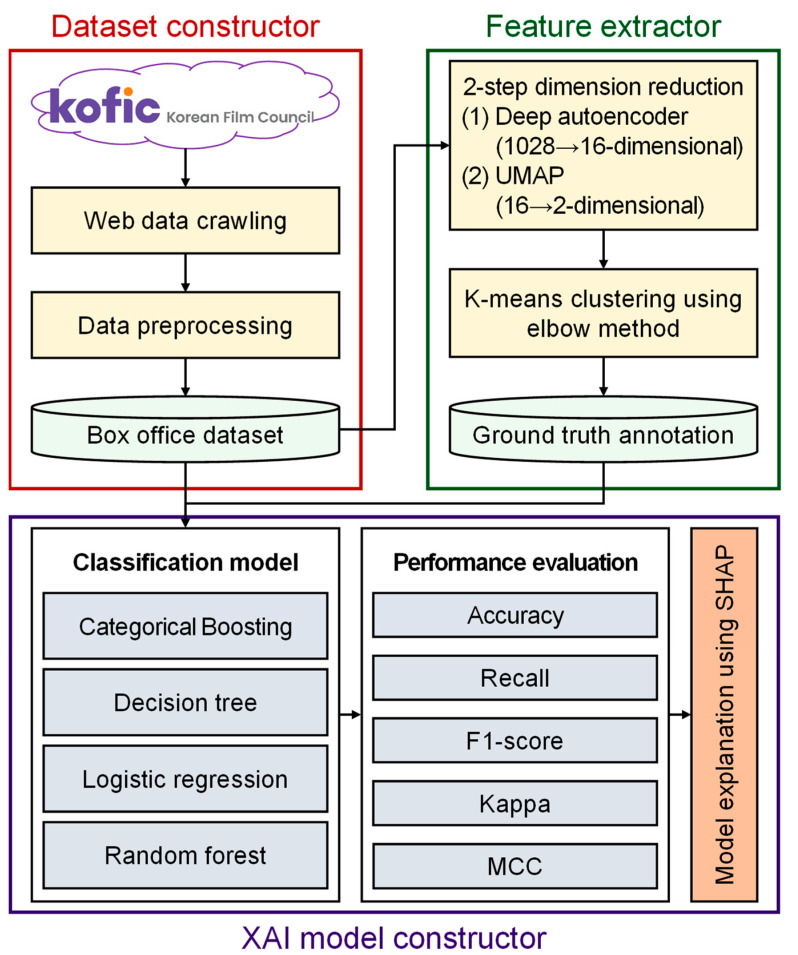
System architecture of the DRECE model.

**Figure 3 entropy-25-00571-f003:**
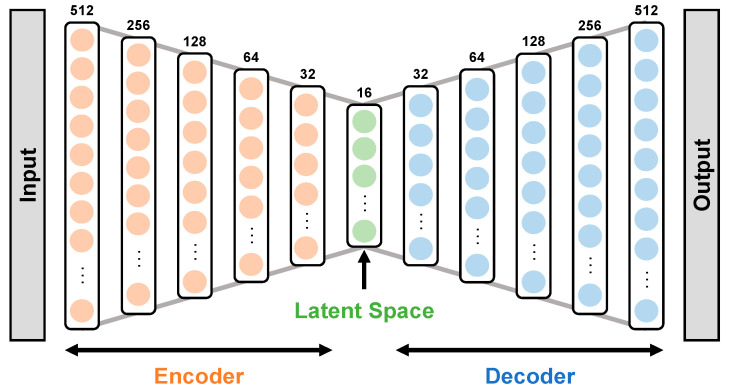
Deep autoencoder architecture.

**Figure 4 entropy-25-00571-f004:**
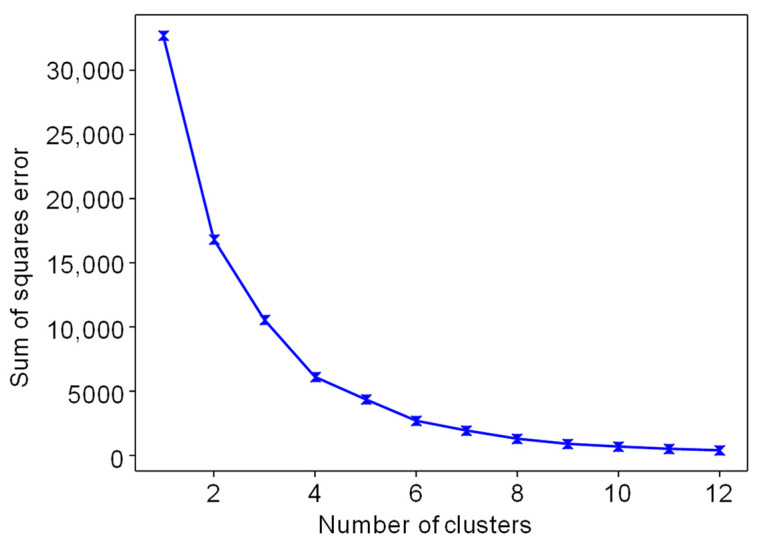
Elbow method for the optimal value of K in K-means clustering.

**Figure 5 entropy-25-00571-f005:**
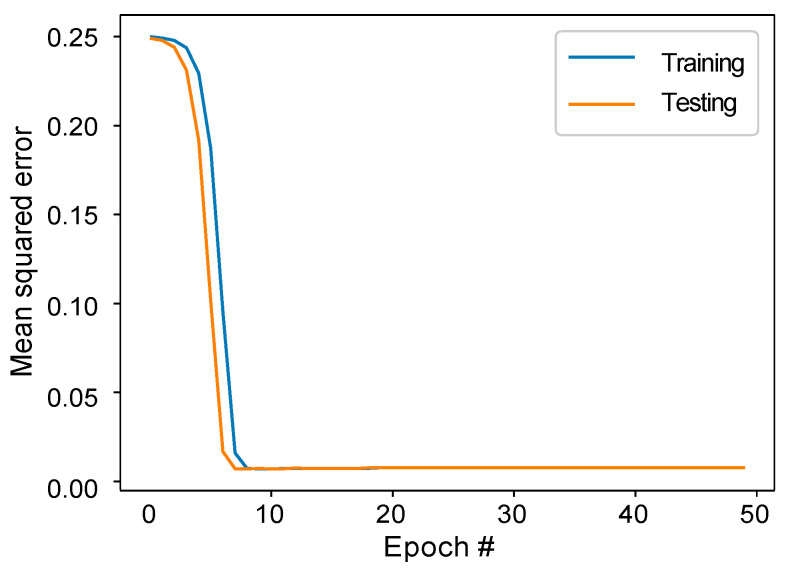
Comparison of training and testing loss for the DAE.

**Figure 6 entropy-25-00571-f006:**
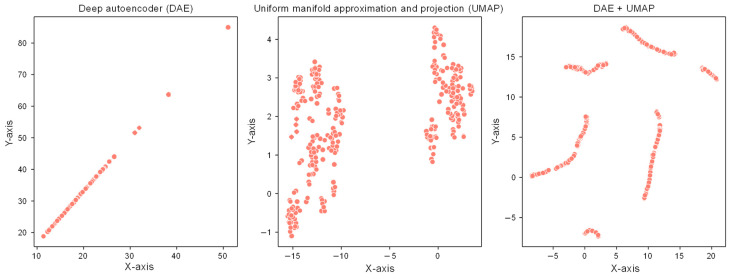
Visualization of dimensionality reduction techniques.

**Figure 7 entropy-25-00571-f007:**
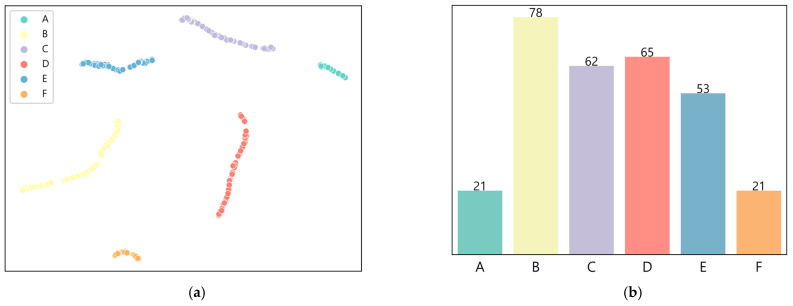
Clustering of box office data into six clusters. (**a**) Result of data clustering into six clusters; (**b**) Number of data points per cluster.

**Figure 8 entropy-25-00571-f008:**
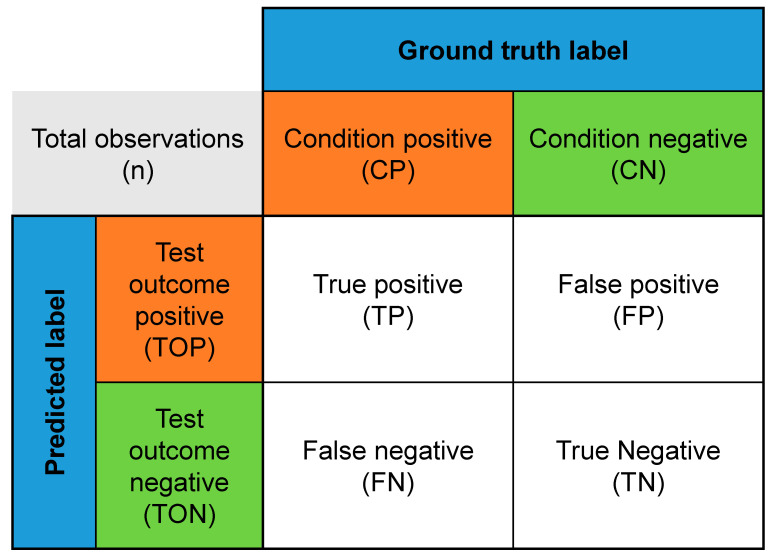
Confusion matrix.

**Table 1 entropy-25-00571-t001:** Summary of AI-based box office prediction and classification models.

Author(s)	Year	AI Techniques	Input Variables (Features)	Output Variable
Zhang et al. [41]	2008	MLBP neural network	Cinema information, competition, content category, nation, propaganda, showing time, star value	Movie class or performance
Kim et al. [42]	2014	GPR, KNN, MLR, SVR	Number of SNS mentions, screening-related information, weekly trends	Box office earnings
Hur et al. [43]	2016	ANN, CART, ISM, SVR	Movie data, viewer sentiments from review text	Number of audiences
Lee et al. [44]	2020	Bagging, Boosting, DT, KNN, linear regression	Movie-related variables, number of eWOMs	Box office at weeks 1, 2, and 3 after release
Lee and Choeh [45]	2020	Bagging, DEA, DT, KNN, linear regression	Four eWOM (i.e., review depth, review rating, review volume, and the number of positive reviews)	Box office revenue
Bogaert et al. [46]	2021	Bagging, DT, GBM, KNN, linear regression, neural network, RF	Movie data (MOV), MGC, and UGC from both Facebook and Twitter	Movie sales data
Pan [47]	2022	ANOVA, regression analysis	Box office, film title, film theme, monthly film box office in 2019, the monthly number of film releases in 2019, number of potential audiences, place of origin, positive comment rate, schedule, score, WOM	Box office revenue
Li and Liu [48]	2022	ARIMA, DNN, linear regression, log-linear regression, ridge regression, RF, SVM	Historical (2002–2010) box office information	China GDP, China NMS, US GDP, US NMS
Ni et al. [49]	2022	Linear regression, stacking (CatBoost, GBM, LightGBM, RF, SVR, and XGBoost)	Baidu search index, China microdata, epidemic, movie attribute	Total box office performance
Velingkar et al. [50]	2022	CatBoost, LightGBM, ridge regression, RF, voting regression, XGBoost	Budget, cast, crew, genres, IMDb ID, IMDb rating, IMDb vote count, MPAA rating, original language, original title, overview, popularity rating, production companies, production countries, release date, revenue, runtime, spoken languages, star power, tagline, TMDb rating, TMDb vote count, title	Box office revenue
Ours		Unsupervised Learning: DAE, UMAP, K-means clustering; Supervised learning: logistic regression, DT, RF, CatBoost; XAI: SHAP	16 genres, director’s name, leading actor’s name, production country	Cluster-specific box office success label

ANOVA, analysis of variance; ARIMA, autoregressive integrated moving average; Bagging, bootstrapped aggregation; CART: classification and regression trees; CatBoost, categorical boosting; DAE, deep autoencoder; DNN, deep neural network; DT, decision tree; eWOM, electronic word-of-mouth; GBM, gradient boosting machine; GPR, gaussian process regression; ISM, independent subspace method; KNN, k-nearest neighbors; LightGBM, light GBM; MLBP, multilayer backpropagation; MLR, multiple linear regression; MGC, mainstream media generated content; MPAA, motion picture association of America; RF, random forest; SHAP, Shapley additive explanations; SVM, support vector machine; SVR, support vector regression; TMDb, The Movie Database; UGC, user-generated content; UMAP, uniform manifold approximation and projection; XAI, explainable artificial intelligence; XGBoost, extreme gradient boosting.

**Table 2 entropy-25-00571-t002:** Information on the movie dataset.

No.	Name	Description	Data Type
1	Title	Movie title	Character
2	Country	Production country	Category
3	Genre	16 genres	Category
4	Director	Director’s name	Character
5	Actor	Leading actor’s name	Character

**Table 3 entropy-25-00571-t003:** Performance comparison of machine learning models.

Metric	Model	Cluster A	Cluster B	Cluster C	Cluster D	Cluster E	Cluster F
Accuracy	CatBoost	0.94	0.76	0.83	0.88	0.83	0.94
	DT	0.96	0.78	0.83	0.85	0.87	0.95
	LR	0.98	0.97	0.96	1.00	0.96	0.94
	RF	1.00	1.00	1.00	1.00	1.00	1.00
Recall	CatBoost	0.10	0.06	0.19	0.46	0.04	0.14
	DT	0.38	0.14	0.23	0.34	0.28	0.29
	LR	0.76	0.88	0.79	1.00	0.77	0.19
	RF	1.00	1.00	1.00	1.00	1.00	1.00
F1-score	CatBoost	0.17	0.12	0.32	0.63	0.07	0.25
	DT	0.55	0.25	0.36	0.50	0.43	0.44
	LR	0.86	0.94	0.88	1.00	0.87	0.32
	RF	1.00	1.00	1.00	1.00	1.00	1.00
Kappa	CatBoost	0.16	0.09	0.28	0.57	0.06	0.24
	DT	0.53	0.20	0.30	0.44	0.38	0.43
	LR	0.86	0.92	0.86	1.00	0.85	0.30
	RF	1.00	1.00	1.00	1.00	1.00	1.00
MCC	CatBoost	0.30	0.22	0.40	0.63	0.18	0.37
	DT	0.60	0.33	0.39	0.52	0.47	0.52
	LR	0.87	0.92	0.87	1.00	0.86	0.42
	RF	1.00	1.00	1.00	1.00	1.00	1.00

## Data Availability

The data presented in this study are available in the Appendix A.

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
