# Peer review of "Towards Data-Driven Decision-Making in the Korean Film Industry: An XAI Model for Box Office Analysis Using Dimension Reduction, Clustering, and Classification"

_entropy, 2023, doi:10.3390/e25040571_

Round 1
Reviewer 1 Report
The subject of this paper is relevant, the methods used to solve the problem of forecasting the demand for films are modern
However, the article has a number of shortcomings:
1. The title of the article is more like a popular article than a research one. I would recommend making the title closer to the text
2. Different abbreviations are entered in the abstract, line 10 contains 2 abbreviations. It is advisable to separate its in the text
3. The analysis of the literature (section Introduction and section 2) does not provide a detailed critical analysis of existing film market research using AI methods, including in Korea. This does not allow to get an idea of ​​the depth of the study. At the same time, after analyzing Table 1, the authors do not give an understanding of how the previous studies differ from the presented in this article. It is not clear, did the authors of the article integrate already known machine learning algorithms to solve the problem of assessing and predicting the demand for a film?
4. Table 3 content is well known. It is desirable to give values estimates of the confusion matrix. Or present the results graphically
5. For the tested RF model (Table 4) all classification accuracy metrics take the value 1. It is necessary to explain the appearance of such values.
Author Response
We would like to extend our sincerest gratitude to Reviewer 1 for their insightful and detailed review. Your comments have been invaluable in improving our work, and we sincerely appreciate the time and effort you have put in to evaluate our submission. Please find the corrected file and our response attached, and thank you again for your support.

Reviewer 2 Report
The paper is concerned with developing an approach dedicated to supporting decision makers in identifying dependencies within the Korean film industry. The authors have proposed a model called the DRECE model that refers to box office classification and trend analysis. Moreover, the authors have presented experimental results that indicate the applicability of the proposed approach. In my opinion, the paper is well written, including literature review, proposed methodology, and numerical experiments. However, the authors could improve the paper in the following issues:
1. The description of clusters A-F in section “Conclusions” (lines 809-819) is unnecessary, because this description is presented in subsection 4.1 (lines 667-744). In my opinion, the description of clusters A-F in section “Conclusions” could be removed.
2. Figures 8-13 could be moved in section “Appendix”, after the main body of the paper and before section “References”.
3. I suggest adding a few sentences in the section “Conclusions” regarding limitations of the proposed approach, and the contribution of this paper to entropy-related issues.
4. Carefully proofreading is needed, because the text includes many mistakes, for example, words “Comaprison” and "Clssification" in the title of subsection 4.2.
Author Response
We would like to extend our sincerest gratitude to Reviewer 2 for their insightful and detailed review. Your comments have been invaluable in improving our work, and we sincerely appreciate the time and effort you have put in to evaluate our submission. Please find the corrected file and our response attached, and thank you again for your support.

Round 2
Reviewer 1 Report
The authors have made adjustments in accordance with the reviewer comments.
The manuscript has been sufficiently improved to warrant publication in Entropy.
Author Response
Thank you very much for your favorable response.